# Effect of distributing locally produced cloth facemasks on COVID-19-like illness and all-cause mortality–a cluster-randomised controlled trial in urban Guinea-Bissau

Line M. Nanque [1,2], Andreas M. Jensen [1,2], Arthur Diness[1], Sebastian Nielsen[1,2], Carlos Cabral[1], Dylan Cawthorne [3,4], Justiniano S. D. Martins[1], Elsi J. C. Ca[1], Kjeld Jensen[3,4], Cesario L. Martins[1], Amabelia Rodrigues[1], Ane B. Fisker [1,2]*

1 Bandim Health Project, INDEPTH Network, Bissau, Guinea-Bissau, 2 Institute of Clinical Research, Bandim Health Project, Research Unit OPEN, Odense University Hospital/ University of Southern Denmark, Odense, Denmark, 3 The Maersk Mc-Kinney Moller Institute, SDU Drone Center, University of Southern Denmark, Odense, Denmark, 4 Engineers Without Borders Denmark, Copenhagen, Denmark

* afisker@health.sdu.dk

**Data Availability Statement:** As part of the data transfer agreement between the National Institute

## Abstract

Facemasks have been employed to mitigate the spread of SARS-CoV-2. The community effect of providing cloth facemasks on COVID-19 morbidity and mortality is unknown. In a cluster randomised trial in urban Bissau, Guinea-Bissau, clusters (geographical areas with an average of 19 houses), were randomised to an intervention or control arm using computer-generated random numbers. Between 20 July 2020 and 22 January 2021, trial participants (aged 10+ years) living in intervention clusters (n = 90) received two 2-layer cloth facemasks, while facemasks were only distributed later in control clusters (n = 91). All participants received information on COVID-19 prevention. Trial participants were followed through a telephone interview for COVID-19-like illness (3+ symptoms), care seeking, and mortality for 4 months. End-of-study home visits ensured full mortality information and distribution of facemasks to the control group. Individual level information on outcomes by trial arm was compared in logistic regression models with generalised estimating equation-based correction for cluster. Facemasks use was mandated. Facemask use in public areas was assessed by direct observation. We enrolled 39,574 trial participants among whom 95% reported exposure to groups of >20 persons and 99% reported facemasks use, with no difference between trial arms. Observed use was substantially lower (~40%) with a 3%, 95%CI: 0–6% absolute difference between control and intervention clusters. Half of those wearing a facemask wore it correctly. Few participants (532, 1.6%) reported COVID-19-like illness; proportions did not differ by trial arm: Odds Ratio (OR) = 0.81, 95%CI: 0.57–1.15. 177 (0.6%) participants reported consultations and COVID-19-like illness (OR = 0.83, 95%CI: 0.56–1.24); 89 participants (0.2%) died (OR = 1.34, 95%CI: 0.89–2.02). Hence, though trial participants were exposed to many people, facemasks were mostly not worn or not worn correctly. Providing facemasks and messages about correct use did not substantially increase their use and had limited impact on morbidity and mortality.

**Trial registration:** clinicaltrials.gov: NCT04471766.

of Public Health (INASA) and the Bandim Health Project, access to data will granted when covered by an ethical permission and approved the data access committee consisting of the director of INASA, the steering board of the Bandim Health Project and key researchers involved in the study. The Guinean Ethics Committee (Comité Nacional de Ética em Pesquisa na Saúde) can be contacted at nabangna.julho2009@gmail.com or djicoblama@gmail.com.

**Funding:** This work was supported by Novo Nordisk Foundation (NNF20SH0064649 to Engineers Without Borders Denmark), Reinholdt W. Jorck Fonden (20-JP-0299 to Engineers Without Borders Denmark), Augustinus fonden (20-1196 to Engineers Without Borders Denmark), COWIfonden (APE/knl/A_149.14 to Engineers Without Borders Denmark) and by University of Southern Denmark (to Bandim Health Project). The funders had no role in study design, data collection and analysis, decision to publish, or preparation of the manuscript.

**Competing interests:** The authors have declared that no competing interests exist.

## Introduction

Facemasks have been one of the tools employed in the attempt to mitigate the spreading of SARS-CoV-2, based on the assumption that they would a) prevent transmission of the infection to others and b) protect the wearer against becoming infected [1]. Furthermore, it has been proposed that facemasks could reduce the inhaled viral load, and thereby reduce disease severity [2]. The use of facemasks is not a stand-alone intervention but has been advised and used in addition to other interventions, such as physical distancing, improved hygiene, and later vaccines.

The guidelines for facemask use have differed across settings and situations. In April 2020, WHO did not recommend the use of facemasks outside health care settings for the general public [3], by June 2020 the WHO guidelines included information on settings where the public could be recommended to wear a non-medical facemasks [4], and by December 2020, the WHO guidelines recommended wearing facemasks indoor, and in settings where physical distancing could not be sustained [5].

During the early phases of the pandemic, a global shortage of medical facemasks limited their availability [6]. Production of reusable facemasks using more readily available materials, such as fabric, was therefore widespread and in many settings recommended [7].

There was little empirical evidence for the effectiveness of medical facemasks or cloth facemasks in community settings. Ten trials that have assessed the effect of facemasks on preventing influenza infections yielded variable results [8] and only observational studies had assessed the effects of facemasks against COVID-19 and other corona viruses [9]. These studies suggested a potential beneficial effect of wearing facemasks in clinical settings, and to a lesser extent, and based on fewer studies, also in a community setting [9]. Thus, in the absence of data from trials, recommendations on facemasks were based on precautionary principles [1] and on extrapolation from efficacy studies.

The permeability of the different facemask materials has been studied [10,11], but the ability of the facemask to protect the wearer and surroundings also depends on the fit of the facemask [12]. Important to the effectiveness is also whether the use of a facemask alters the behaviour of the wearer and the surroundings: It has been hypothesised both that the use of facemasks could affect the behaviour of carriers and their surroundings making them more careful [13] or take more risks [14], and that facemask use would increase or decrease the risk of transmitting virus via fomites due to increased/decreased touching of the face [15].

Reusable cloth facemasks constitute a low-cost and sustainable preventive measure, which could potentially help mitigate the impact of COVID-19, especially in resource constrained settings where physical distancing is particularly difficult. However, so far, no studies have assessed the effect of facemask use on preventing infectious transmission in an African setting. We conducted a cluster-randomised trial to assess the impact of providing two 2-layer cloth facemasks and messages on preventive measures versus messages only for all individuals over the age of 10 years in an urban African setting during the COVID-19-pandemic. The outcomes were self-reported COVID-19-like illness, health care contacts for COVID-19-like illness and all-cause mortality during 4 months of follow-up.

## Materials and methods

### Setting

The facemask trial was conducted using the Bandim Health Project's (BHP) urban Health and Demographic Surveillance System (HDSS) in Guinea-Bissau. Since 1978, the BHP has monitored the health and survival of the population in suburbs of the Guinean capital, Bissau.

Today the urban BHP HDSS covers six districts with 37 zones and 324 geographically defined clusters (mean number of houses/cluster: 19) and a population of approximately 100,000 individuals of whom 73,500 are over 10 years of age (Fig A in S1 Text).

The first case of SARS-CoV2 infection in Guinea-Bissau was registered on March 25, 2020. When we initiated the trial in July 2020, a total of 1949 cases had been detected by PCR-test which were the only tests available. During the period of enrolment, the number increased to 2510 and 4 months later it was 3751 [16]. However, only a small proportion of the SARS-CoV-2 infections were likely detected: among 140 staff at BHP who were tested for antibodies in November 2020, 18% were IgG positive, but only 3 of the 25 persons with serological evidence of prior SARS-CoV2 infection reported having had a PCR test performed [17]. Facemask-use was mandated in Guinea-Bissau throughout the trial starting from May 11, 2020 [18] with the police intermittently enforcing the use on streets and in market areas [19,20]. Curfews were implemented and enforced to different extents from March 2020 until 26 June 2020 [21] and had thus ceased before trial start. COVID-19 vaccination in Guinea-Bissau started in April 2021 [22].

## Study design and randomisation

The facemask trial was conducted in 20 zones with 182 clusters in three of the BHP districts: Cuntum-II, Bandim-I and Bandim-II. Clusters were randomised 1:1 to the intervention or the control group stratified by zone, using a computer algorithm to draw a random combination of clusters. To ensure balanced groups regarding key co-variates (population size, age distribution and households with functioning electricity), we used covariate-constrained randomisation (Supplementary Methods in S1 Text). The trial was unblinded.

All households in the included study area were visited during the day. A list of all residents ≥10 years of age was extracted from the BHP database. Field assistants verified and updated the list with new residents, migrations, and deaths. All household members aged ≥10 years in the study areas were eligible for enrolment. Provided oral consent, household members were enrolled and the enrolling field assistant documented consent on the enrolment form. If any household member ≥10 years of age was temporarily absent, a relative could accept enrolment on their behalf. For individuals under 18 years of age, we asked for a guardian's oral consent. The oral consent was used to maintain physical distance and described in the study protocol submitted for ethics evaluation.

During the visit, we updated the information on the socioeconomic variables related to household characteristics (type of roof, porch, source of water supply) and collected up to two telephone numbers for each household member in both the intervention and control group. In addition, we updated the information on sleeping arrangements (room and bed for each person) and whether the household had an indoor ceiling presumed to be a risk factors for infection in multi-family houses [23].

## Intervention

Intervention and control households were provided with the key messages given in the country for prevention of COVID-19. These messages included frequent hand washing with soap, sanitizer or bleach solution, physical distancing (1 m apart), and avoiding crowds, hand shaking and kissing. The messages were short and communicated with pictures (Fig C in S1 Text).

Upon enrolment, each study participant in the intervention households received two cloth facemasks. The field assistant instructed the participants to use them each time they went out or when they were in close contact with many individuals not from the household. Furthermore, intervention group participants were instructed to wash the facemask with warm water

and soap before every use, and not to touch the facemask (Fig C in S1 Text). No facemasks were provided to participants in the control group at enrolment; they received one cloth face-mask at the end of the study (Fig D in S1 Text). Group allocation was not concealed at enrolment.

The facemasks were locally produced in four sizes (small, medium, large, extra-large) based on a model developed at the University of Southern Denmark (Supplementary Methods in S1 Text). The facemasks had two layers of woven cotton fabric and a metal wire at the nose bridge to maintain a close fit. They were held in place by two horizontal elastic straps, one worn across the back of the head and the other across the back of the neck. When worn correctly, the facemask covered the lower half of the face from the nasal root to under the chin and jaw (Fig E in S1 Text) and provided a close fit even during movement. Based on laboratory measurements, the facemask cloth fabric was found to filter 19% of 0.4 μm particles (salt) and 11% of 0.6 μm particles (paraffin oil). It should be noted that the size of aerosols of a sneeze containing SARS-CoV-2 is larger [10] (~3 μm; Supplementary Methods in S1 Text).

**Assessment of outcomes.** Participants were followed up through telephone calls 4 months after enrolment. We had intended more frequent follow-up calls (every 2 months) but were unable to implement this for all participants for logistical reasons (Supplementary Methods in S1 Text).

During the telephone follow-up calls, individual-level information was collected on the four prespecified primary outcomes; self-reported COVID-19-like illness (defined as three or more of the following symptoms of COVID-19: fever, cough, fatigue, shortness of breath, loss of sense of smell/taste) and/or reported positive test (PCR-test performed by health personnel), consultation with COVID-19-like illness, severe COVID-19-like illness (defined as hospitalisation or death), and all-cause mortality. Information on these outcomes was obtained by asking the participant or guardian (for children <18 years of age), whether the participant, since enrolment, had experienced any episodes of illness, including any of the listed COVID-19 symptoms and/or other symptoms and had sought consultation, been hospitalised, or died. If the participant was absent, relatives in the same compound or neighbours were interviewed. Except for self-reported COVID-19-like illness, for which we had not collected information on date of the episode, only events occurring within 4 months (121 days) of enrolment were included in the analyses.

Information about potential exposure to SARS-CoV-2 was collected at the telephone interviews by asking whether participants attended any event(s) with 20 or more people not living with the participant or had been in contact with a person with COVID-19. Information on past exposure was not collected for individuals who had died. For the compliance sub-study, a field assistant was stationed in different clusters (outside, 2 hours of observation per session) to observe facemask use by counting the number of people in the vicinity and among those, the number of people wearing a facemask (Supplementary Methods in S1 Text).

All participants received a home visit after the telephone call after 4 months of follow-up to ensure that we had complete information on vital status for all enrolled participants, and to distribute facemasks to participants in the control group (Fig D in S1 Text).

## Sample size

With an anticipated 90% acceptance rate, we expected to be able to enrol around 19,290 in each randomisation group in the study area. Based on the estimated number of enrolments within each cluster and assuming a baseline mortality risk of 1%, and a uniform mortality distribution of 0.5–1.5%, we estimated an 80% power to demonstrate an effect of the intervention if the true mortality reduction by the intervention was 27% (alpha = 0.05). For more frequent

outcomes we would have power to show smaller effects: for example, for illness (anticipated baseline rate 10%, uniform variation 5–15%), we estimated an 80% power to show a reduction if the real effect was a 10% reduction.

## Statistical analyses

**Primary analysis.** Analyses were based on individual level data. We compared the proportion of individuals reporting a study outcome by group allocation using logistic regression models with generalised estimating equation (GEE)-based correction for cluster, robust standard error, and an exchangeable correlation matrix. We adjusted the estimates for zone (as the randomisation was stratified by zone) and the balancing variables: number of people living in the cluster, proportion of individuals aged ≥50 years and proportion of households with functioning electricity (Supplementary Methods in S1 Text). No adjustment was made for multiple testing. All analyses were made on the study population receiving facemasks according to their randomisation arm. Observations with missing information on the outcome were excluded from the analyses.

**Secondary analyses.** We investigated whether the effect of facemask distribution varied by potential effect modifiers identified as important for the severity of COVID-19, exposure to SARS-CoV-2 or for the effect of facemasks in prior studies; sex [24,25], age [26], and having an indoor ceiling (yes/no), attending events with ≥20 persons, reported contact with a person with COVID-19, number of children <10 years registered in household, and number of people sleeping in the same room. We compared self-reported and observed facemask compliance by group.

**Sensitivity analyses.** For the morbidity outcomes (COVID-19-like illness and consultation), we investigated whether the proportions differed for participants who received two telephone follow-up interviews (at both 2 and 4 months after enrolment) and those who received only one follow-up call. We did so by including an 'intensity of follow-up'-variable as a potential effect modifier.

Furthermore, due to delays in implementing the follow-up visits, we investigated whether extending the period of follow-up from 4 months to the date of the telephone interview/home visit affected the conclusions. During 2020 and 2021, Guinea-Bissau experienced three waves of COVD-19 infections during which the risk of COVID-19 was higher (Fig D in S1 Text); 1) before 1 October 2020, 2) 1 January—1 April 2021, and 3) after 1 July 2021. We generated a continuous measure of the proportion of follow-up time within these high-exposure periods and explored whether the effect varied by exposure.

In explorative analyses, we assessed whether conclusions differed when the morbidity analyses were limited to the observations where a) the participant provided information themselves, b) the participant was present at enrolment, and c) the participant was registered in the HDSS before enrolment. Furthermore, for reported morbidity outcomes, we assessed the impact of adjusting for time elapsed between enrolment and first successful follow-up call (number of days after 6 weeks). Finally, to compare mortality registered during the trial with the pre-trial mortality, we estimated the crude annual mortality rate among persons aged 10 + years under surveillance in the same zones in 2013–2019.

## Ethics

Our trial protocol was approved by the Guinean National Ethics Committee (Comité Nacional de Ética na Saúde, reference: 79/CNES/INASA/2020). The explanations to participants included information on social distancing and hygiene and it was stressed that wearing a facemask, was complementary to other measures of protection. BHP field assistant used individual

protection (facemask, hand sanitizer) and maintained physical distance. Enrolment was initiated in parallel with the production of facemasks; the production pace was the limiting factor for enrolment and facemask distribution.

### Role of the funding source

The funders had no role in study design, data collection, analysis, writing of the report or decision to submit the paper for publication.

## Results

Between 20 July 2020 and 22 January 2021, we enrolled 39,688 trial participants among 39,721 eligible residents in the study area (Fig 1). In total, 8712 households in 181 of the eligible 182 clusters took part in the trial; 90 clusters in intervention arm and 91 in in the control arm. Enrolment in intervention and control clusters took place in parallel (Fig G in S1 Text). The median age at enrolment was 27 years. Most enrolled people shared a bedroom with at least one other trial participant. Though some of the background factors differed statistically significantly between the trial arms, the absolute differences were small (Table 1). However, more new residents were registered at the enrolment visit in the intervention arm (2126) than in the control arm (1447) (Fig 1, Supplementary Results in S1 Text).

114 persons were excluded due to protocol violations (control group receiving facemask: 92, intervention group not receiving facemask: 22) retaining 39,574 persons to be followed (Fig 1). We made 60,677 telephone calls and succeeded in obtaining information through telephone calls for 33,306 persons (84%; intervention: 16,779 (84%), control: 16,527 (84%) (Fig 1, Table A in S1 Text)). The median time elapsed between enrolment and the first successful call was 227 days, interquartile range (IQR): 104–246 with slightly shorter intervals in the intervention (210 days, IQR: 85–241) than control arm (231 days, IQR: 150–250). For 37% of the successful calls, the trial participants provided information for themselves (intervention: 39%, control 36%; $p < 0.001$) (Table A in S1 Text). Among the trial participants with successful follow-up calls, 5% (1518/33,306) reported to have had a COVID-19 test performed, but only 24 of these tests (1.6%) were reported positive.

All individuals were also visited at home to ensure full information on survival and to confirm deaths registered through the telephone interviews. Ninety-seven percent (intervention 97%; control 98%) were still registered to be living in the study area 121 days after enrolment and 7 people (intervention: 3; control: 4) were lost to follow-up (Fig 1).

Most participants (95%) reported exposure to large groups of people (events with $\geq 20$ people from outside their household) and the majority (90%) reported the events as frequent/recurring, such as attending school, going to the marked or going to work. The absolute differences between the intervention and control groups were small (Table A in S1 Text).

In both groups, 92% reported to use a facemask whenever leaving home. An additional 7% reported facemask use sometimes. The type of facemask used differed by study arm: 55% of the intervention group (63% among those providing information on facemask type) reported wearing cloth facemasks, while this was 22% (33%) in the control group. In contrast, the use of a medical facemasks was more common in the control group, 44% (among those with information on type: 67%) than in the intervention group 36% (37%) (Fig 2 and Table A in S1 Text). In the compliance sub-study, the observed facemask use was much lower than reported through the telephone follow-up: only 40% of individuals were observed to wear a facemask with little difference between observations made in the intervention and control areas (Fig 3). Adjusting for calendar time, an additional 3%, 95%CI: 0–6% of people observed in the intervention zones wore facemasks (Supplementary Results in S1 Text). Only half of the facemask

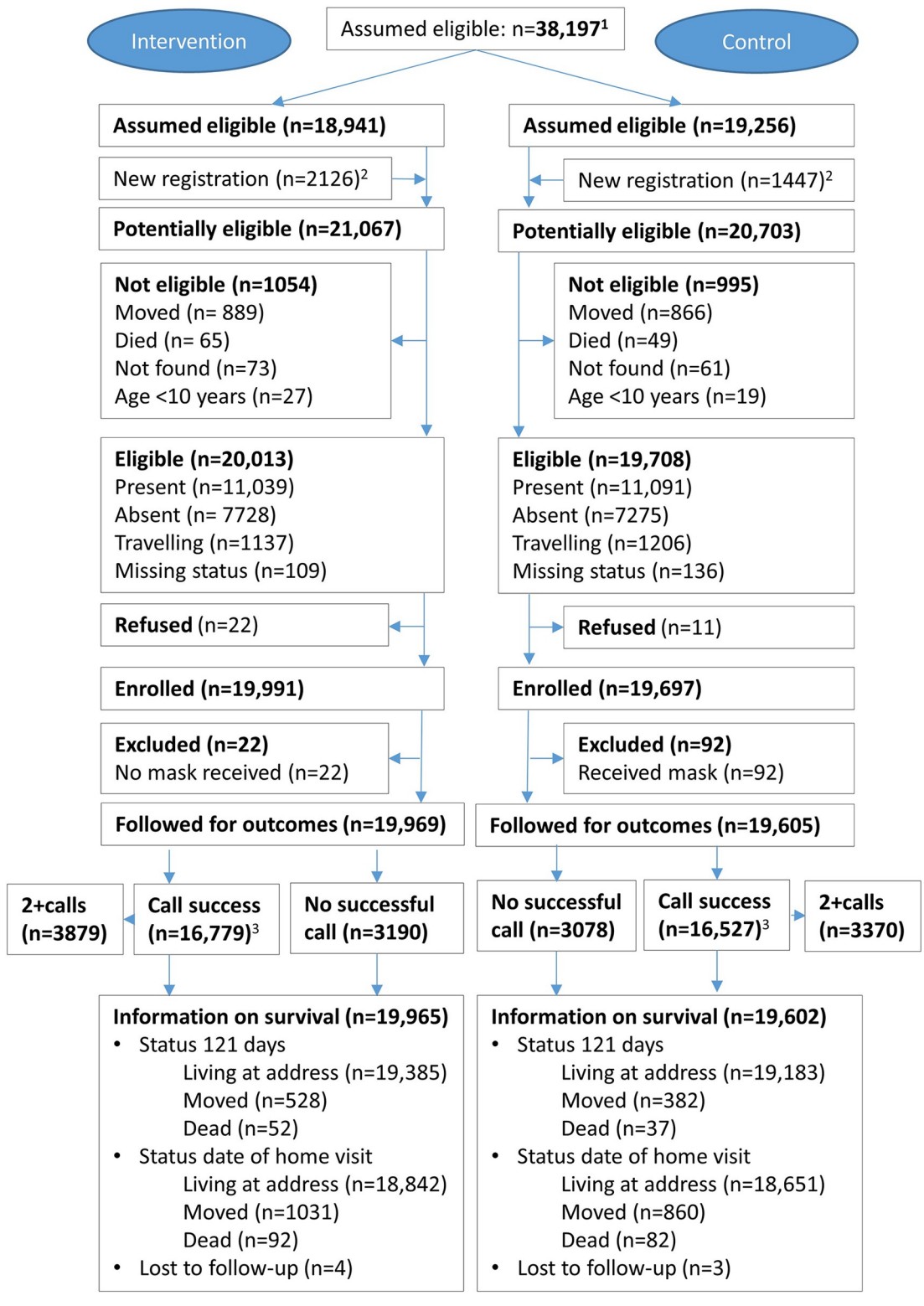

**Fig 1. Participant flow in the Guinea-Bissau Face Mask trial.** Notes: 1: Registered in the Bandim Health and Demographic Surveillance System before visit to enrol; 2: Added to population list during enrolment visit; 3: Participants with successful telephone calls contributing to the analysis for the main outcome "COVID-19-like-illness".

**Table 1. Baseline characteristics according to group allocation.**

| | Intervention [n, (%)] | Control [n, (%)] | P-value, test of same distribution[a] |
|---|---|---|---|
| Number | 19969 (50) | 19605 (50) | |
| Number of households | 4428 (51) | 4284 (49) | |
| Males | 9384 (47) | 9224 (47) | 0.91 |
| Age (years), median (IQR)[b] | 27 (19–38) | 27 (19–39) | 0.44 |
| **District** | | | 0.01 |
| Bandim-I | 10477 (52) | 10465 (53) | |
| Bandim-II | 4794 (24) | 4452 (23) | |
| Cuntum-II | 4698 (24) | 4688 (24) | |
| Number of people sleeping in same room, median (IQR) | 2 (2–3) | 2 (2–3) | 0.001 |
| **Socio-economic factors** (compared on household level) | | | |
| Type of roof[c] | | | 0.08 |
| Straw | 14 (0) | 18 (0) | |
| Zink | 4221 (98) | 4128 (98) | |
| Tiles | 58 (1) | 36 (1) | |
| Cement | 10 (0) | 17 (0) | |
| Other | 1 (0) | 0 (0) | |
| Type of veranda[c] | | | 0.31 |
| Mosaic | 466 (11) | 447 (11) | |
| Cement | 3636 (85) | 3595 (86) | |
| Stamped soil | 184 (4) | 154 (4) | |
| Has ceiling[d] | 1478 (35) | 1437 (35) | 0.77 |
| Type of water source[c] | | | <0.001 |
| Tap inside | 773 (18) | 699 (17) | |
| Tap outside | 2723 (64) | 2841 (68) | |
| Small private well | 431 (10) | 308 (7) | |
| Public well | 331 (8) | 311 (7) | |
| Functioning electricity in household (family-level)[e] | 629 (61) | 670 (63) | 0.25 |
| **Intention to use mask**[c] | | | |
| Yes | 11,400 (58) | | |
| No | 9 (0) | | |
| Don't know | 8380 (42) | | |
| **Intention of when to use mask**[c] | | | |
| When leaving house | 8775 (78) | | |
| Often | 2426 (22) | | |
| **Reason for mask use refusal**[c] | | | |
| Not able to wear a mask | 2 (29) | | |
| Prefers medical mask | 4 (57) | | |
| Can't explain why | 1 (14) | | |

a: Test of same distribution using chi2 for categorical variables and rank-sum tests for continuous variables

b: One person (control) with missing information on age

c: Numbers do not add up due to some having missing information

d: No information on ceiling for 305 households (189 intervention, 126 control)

e: Accessed for households with a registered birth in the last 5 years, thus missing information for 6619 households (3394 intervention; 3225 control).

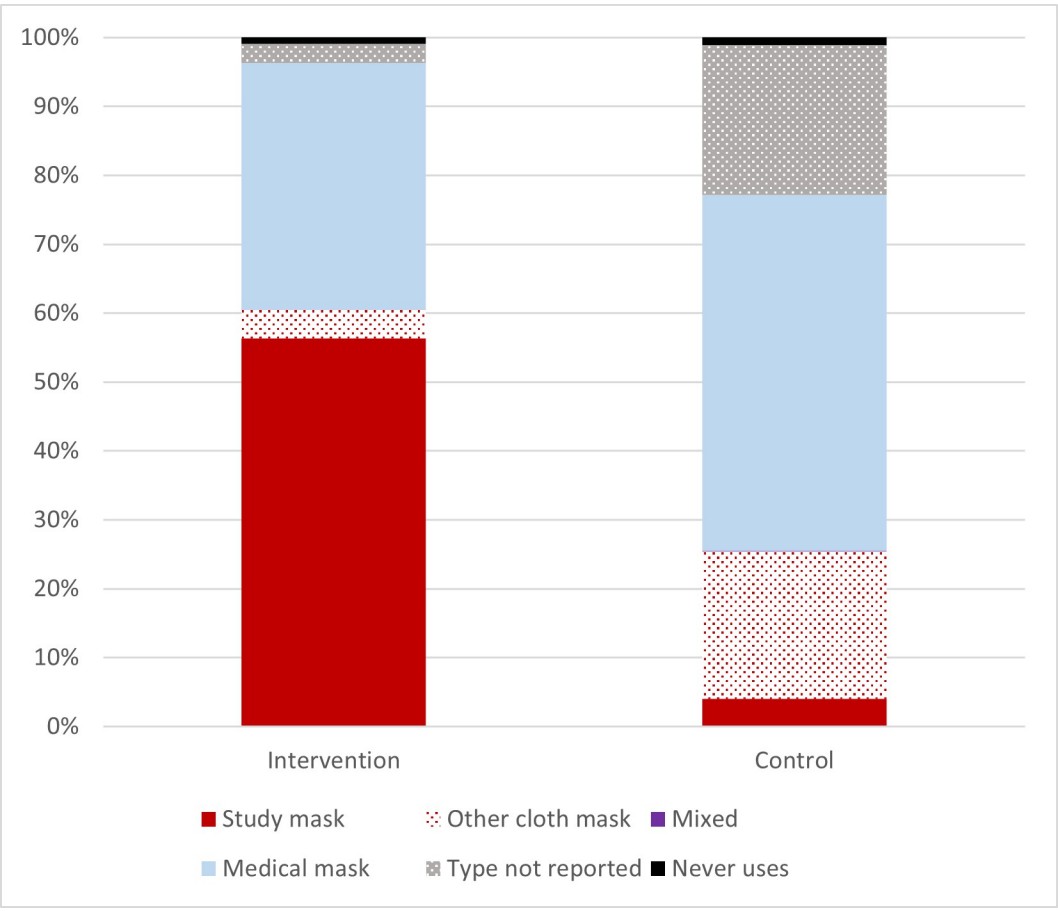

**Fig 2. Mask use reported through telephone follow-up.**

wearers wore facemasks which covered both nose and mouth (Supplementary Results and Fig H in S1 Text).

Among the 33,406 participants followed-up through the telephone interviews there were a total of 532 episodes of COVID-19-like illness (intervention: 244 (1.45%); control: 288 (1.74%)). There was no statistically significant difference by group: OR: 0.81, 95%CI: 0.57–1.15 (Fig 4 and Table 2). Every third episode of COVID-19-like illness (intervention: 82 (0.51%); control: 95 (0.60%)) was accompanied by a consultation within the 4 months of follow-up, with no difference between the trial arms resulting in an OR of 0.83, 95%CI: 0.56–1.24 (Fig 4 and Table 2). We registered only 11 hospital admission within 4 months of follow-up among participants with COVID-19-like illness (intervention: 8, control: 3; OR = 3.21, 95%CI: 1.10–9.39). Only one of the admissions (intervention) was reported to have been diagnosed as COVID-19 (Supplementary Results in S1 Text). There were 89 deaths during the 4 months of follow-up, 52 (0.26%) in the intervention group and 37 in the control group (0.19%), OR = 1.34, 95%CI: 0.89–2.02 (Fig 4 and Table 2).

The analyses of the investigated effect modifiers did not indicate that a potential effect varied significantly by sex or age (Table 2, Fig 4 and Table B in S1 Text). Nor did the effect differ by factors potentially associated with COVID-19 exposure and there was no consistent pattern across the different indicators of exposure (Table B in S1 Text).

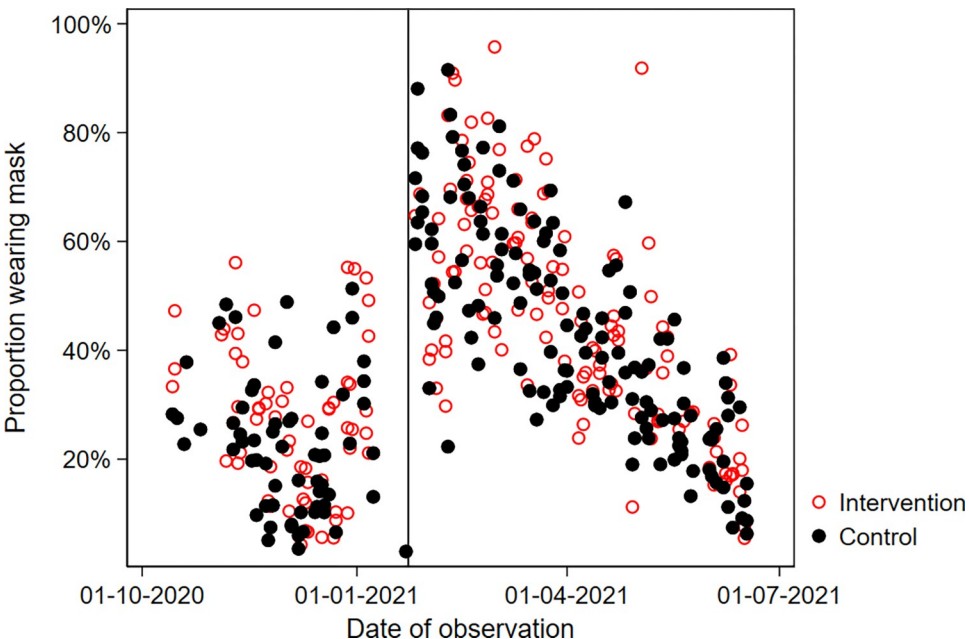

**Fig 3. Direct observations.** Proportion of individuals observed wearing a face mask during each 2-hour observation session. Note: Vertical line indicates January 22, 2021, when a sudden change in observed facemask use occurred.

Extending the follow-up period for consultations with COVID-19-like symptoms to the time of the telephone interview, half the reported episodes with COVID-19-like illness led to consultation. The OR was 0.84, 95%CI: 0.56–1.25 overall, 0.97, 95%CI: 0.61–1.56 for men and 0.70, 95%CI: 0.43–1.13 for women (Table C in S1 Text). Extending the mortality analysis to

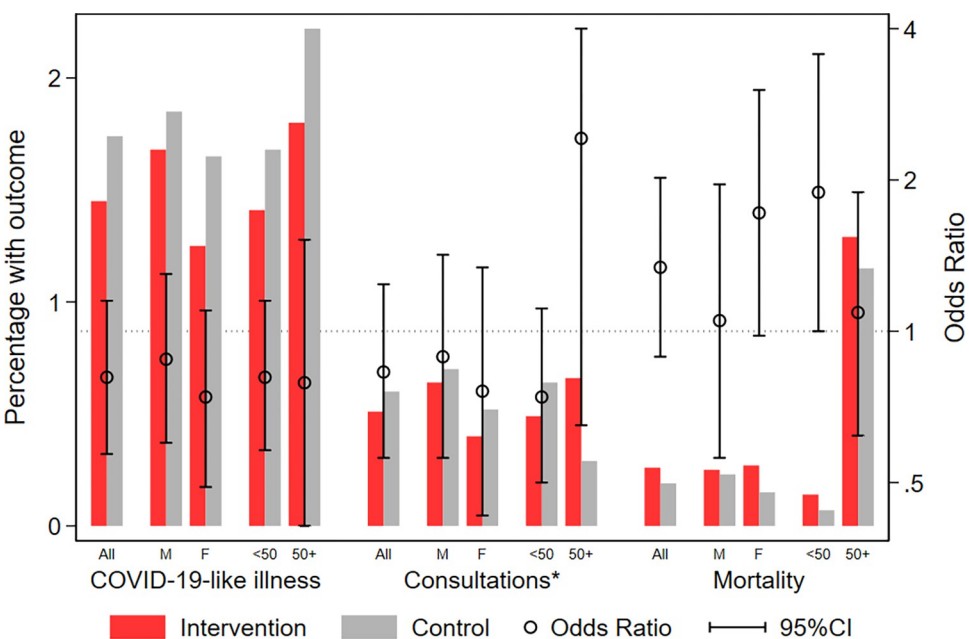

**Fig 4. Effect of providing cloth face masks on self-reported COVID-19-like illness, consultations and mortality.** Note: *Concurrent report of consultations and COVID-19-like illness.

**Table 2. Effect of providing cloth face masks on self-reported Covid-19-like illness, consultations and mortality.**

| | | Intervention [% (events/n)] | Control [% (events/n)] | OR (95%CI)[a] | P-value test of same effect across strata |
|---|---|---|---|---|---|
| **Self-reported COVID-19-like illness** | | | | | |
| Overall | | 1.45 (244/16779) | 1.74 (288/16527) | 0.81 (0.57–1.15) | |
| Sex | Male | 1.68 (132/7837) | 1.85 (143/7748) | 0.88 (0.60–1.30) | 0.34 |
| | Female | 1.25 (112/8942) | 1.65 (145/8779) | 0.74 (0.49–1.10) | |
| Age | <50 years | 1.41 (213/15056) | 1.68 (248/14724) | 0.81 (0.58–1.15) | 0.90 |
| | > = 50 years | 1.80 (31/1723) | 2.22 (40/1803) | 0.79 (0.41–1.52) | |
| **Health centre/outpatient consultations for COVID-19-like illness during 4 months of follow-up** | | | | | |
| Overall | | 0.51 (82/16009) | 0.60 (95/15764) | 0.83 (0.56–1.24) | |
| Sex | Male | 0.64 (48/7507) | 0.70 (52/7439) | 0.89 (0.56–1.42) | 0.63 |
| | Female | 0.40 (34/8502) | 0.52 (43/8325) | 0.76 (0.43–1.34) | |
| Age | <50 years | 0.49 (71/14354) | 0.64 (90/14041) | 0.74 (0.50–1.11) | 0.08 |
| | > = 50 years | 0.66 (11/1655) | 0.29 (5/1723) | 2.42 (0.65–9.02) | |
| **All-cause mortality during 4 months of follow-up** | | | | | |
| Overall | | 0.26 (52/19965) | 0.19 (37/19602) | 1.34 (0.89–2.02) | |
| Sex | Male | 0.25 (23/9382) | 0.23 (21/9224) | 1.05 (0.56–1.96) | 0.26 |
| | Female | 0.27 (29/10583) | 0.15 (16/10378) | 1.72 (0.98–3.02) | |
| Age | <50 years | 0.14 (26/17957) | 0.07 (13/17523) | 1.89 (1.00–3.56) | 0.20 |
| | > = 50 years | 1.29 (26/2008) | 1.15 (24/2079) | 1.09 (0.62–1.89) | |

a: Estimated in logistic regression models with generalised estimating equation-based correction for cluster. Adjusted for zone and the cluster balancing variables: number of people, proportion aged ≥50 years and proportion of households with functioning electricity.

the date of the home visit, there were 92 deaths in the intervention arm and 82 in the control arm, the OR for the intervention/control groups being 1.09, 95%CI: 0.83–1.43 overall, 0.86, 95%CI: 0.58–1.26 for men and 1.43, 95%CI: 0.93–2.18 for women (Table C in S1 Text).

The proportions reporting COVID-19-like illness and consultations were substantially higher for participants who were called two or more times, than for participants called only once during the follow-up period: COVID-19-like illness: ~5 times as high; consultations: 4–10 times as high dependent on period of assessment (Table D in S1 Text). The effects of face-mask distribution did not differ by follow-up intensity, and we observed the same tendency of a beneficial effect for women but not for men (Table D in S1 Text).

The 37% percent of trial participants who provided information on their own behalf during the telephone follow-up, reported 79% (418/532) of the episodes with COVID-19-like illness and 75% (132/177) of the consultations within 4 months of enrolment. However, limiting the morbidity analyses to this group meant little for the estimated effects overall or by sex (Table E in S1 Text).

The proportion of follow-up time during the higher SARS-CoV2-transmission periods varied from 24–74% during the period of enrolment (Fig J in S1 Text). The effect did not markedly differ by whether more or less than 50% of follow-up time was in high transmission periods (Table F in S1 Text).

The explorative analyses limiting the analysis to participants who were registered in the HDSS prior to the enrolment visit and to participants who were present on the date of enrolment did not alter conclusions (Table G in S1 Text). Adjusting for time elapsed between enrolment and first successful follow-up call gave very similar results: COVID-19-like illness, OR = 0.81, 95%CI: 0.57–1.16; consultation: OR = 0.84, 95%CI: 0.56–1.25.

The crude pre-trial mortality in the same zones and age group was 7.50 per 1000 person years (Supplementary Results in S1 Text).

## Discussion

In this first trial of the effect of facemask distribution in an African setting, we found no statistically significant effect on the proportion reporting COVID-19-like illness, seeking consultations for illness or mortality. However, we also found that distributing facemasks and providing information on their use was insufficient to ensure high facemask use. The proportions reporting COVID-19-like morbidity and consultations were much lower than anticipated, and we found indications that we have substantially underestimated the morbidity burden. Nevertheless, even for mortality, where we are confident that we have obtained full information, the rates were much lower than anticipated, potentially indicating that the impact of COVID-19 on the population was less than feared.

By performing the trial in the context of the BHP HDSS, we ensured that we could follow participants after enrolment, limiting the loss to follow-up. Measuring several outcomes does however increase the risk of detecting spurious associations. By cluster-randomising, we aimed to be able to measure the effect on a population level. However, clusters were very close to each other (Fig A in S1 Text), and even if distributing facemasks had greatly reduced the risk of transmission, that effect would also affect the neighbouring clusters. Indeed, in the direct observations performed at public spaces situated in or at the edges of the clusters, we observed little difference between the use of trial facemasks. Thus, people observed passing through the areas were likely from a mix of intervention and control clusters, and hence measuring a potential differential use becomes problematic. Furthermore, in contrast to when we planned the trial, facemask use was mandated in Guinea-Bissau throughout the trial period. Thus, the information on use obtained through the telephone follow-up is likely influenced by social desirability bias. Nevertheless, the facemask mandates likely increased the mask-use in both the intervention and control group and may have meant that we have been comparing an intervention group who (at times) used the provided cloth facemasks, with a control group who were more likely to use medical facemasks. However, it should be noted that medical facemasks were not worn as single-use facemasks but often reused, presumably due to the high cost [27].

While we have high quality and complete information on mortality, we detected very few cases of COVID-19-like illness, with only 1.5% of participants reporting an episode. Our analyses showed higher proportions of events when called with a shorter recall period, hence the recall periods were clearly too long to provide a full picture. Furthermore, a poorer recall on behalf of other household members (~5 times lower proportion), was also evident. Nevertheless, as the conclusions did not change when restricting the analyses to information being provided by the participant or adjusting for recall period, this should not affect the relative comparison.

COVID-19-like illness was based on symptoms compatible with SARS-CoV2 infection, but the symptoms overlap with other infections. Hence, in addition to the low sensitivity described above, the specificity is not likely to be high: In Bangladesh, approximately 26% of persons with recent symptoms of COVID-19-like illness had serological evidence of past infection [26]. Nevertheless, a large proportion of the Guinean population has likely had COVID-19 [17].

Two other randomised trials have now assessed the effects of wearing facemasks. In Denmark, an individually randomised trial, hence not designed to assess the effect of facemasks on transmission, showed no overall effect of facemasks on the proportion having a SARS-CoV-2 infection during 1 month of follow-up (OR = 0.82, 95%CI: 0.54–1.23) [28]. However, a potential beneficial effect may have been present for women (OR = 0.65, 95%CI: 0.38–1.12) but not for men (OR = 1.12, 95%CI: 0.59–2.12) [28]. Self-reported mask-use (predominantly or fully as recommended) in the intervention group was 93%, but there was no observation of actual use [28]. In Bangladesh, 600 villages with a population of 342,183 were randomised to an

intervention or control group. In the intervention group, facemasks (surgical or cloth) were distributed and interventions to increase facemask use during the 8-week follow-up period were implemented. The combined intervention increased observed facemask wearing from 13% to 42% while facemask use was promoted but it dropped to baseline levels 3 months after the facemask use promoting activities stopped [26]. There was an 11.6%, 95%CI: 6.6–16.6% lower incidence of COVID-19-like illness during follow-up and during the end of trial survey, 9.5%, 95%CI: 0.5–18.5% fewer participants in the intervention group (0.68%) reported episodes of COVID-19-like illness and had COVID-19 antibodies than in the control group (0.76%) [26]. The Bangladeshi trial reports stronger benefits for medical facemasks than for cloth facemasks, however facemask type appeared to be less important for women (who also wore facemasks more frequently) than for men. Thus, the Bangladeshi trial showed a COVID-19-protective effect of facemasks at the community level.

Mortality was not assessed in the trial in Bangladesh [26]. Though mortality in our trial tended to be higher in the intervention than in control group (OR = 1.34, 95%CI: 0.89–2.02 with 4 months of follow-up and OR = 1.09, 95%CI: 0.83–1.43 with the extended follow-up), it should be noted that mortality registered at the enrolment visits also tended to be higher in the intervention than in the control group (Fig 1 and Supplementary Results in S1 Text). Nevertheless, the overall mortality was not higher than during the prior years. The 89 deaths among the 39,657 participants translates to approximately 6.7 per 1000 person years, while annual mortality among individuals in the same area in 2013–19 was 7.5 per 1000 person years. There is little empirical data on all-cause mortality in Africa during the pandemic [29]. Hence most African mortality estimates are based on models [29,30].

Substantially higher reported than observed facemask use also been seen in Kenya [31] and in Bangladesh [26], limiting the usefulness of information on reported use.

Providing free cloth facemasks to the study population in a context with mandated facemask use did not increase the reported or observed facemask wearing substantially. While our results are compatible with reductions in COVID-19-like illness, the confidence intervals are wide. As indicated by the secondary and explorative analyses, our results are also compatible with a potential benefit on mask distribution on morbidity for females. It could be speculated that women are more compliant with facemask use recommendations, have different exposure intensities or that different infectious doses are necessary to cause illness. However, the under-reporting impairs our ability to measure a potential effect. Furthermore, according to the information reported during the follow-up calls, the control group was more likely to wear a medical facemask, which may have provided better protection against COVID-19 [26]. If anything, the indicators of severe morbidity, the hospitalisation and mortality data, indicated that the risk was higher in the intervention group.

The facemask trial was conducted prior to vaccines being available in Guinea-Bissau and the preventive measures included physical distancing, avoiding large groups of people and improved hygiene. As indicated by the follow-up calls, compliance with the advised avoidance of crowds was low. Such recommendations are difficult to follow in a setting where at least one household member must go to the market every day and working from home is not possible. Thus, participants were presumably widely exposed to SARS-CoV-2. However, as interactions take place outside buildings, people are exposed to lower virus doses. This supports that the partial protection from wearing a facemask could be clinically important [32].

There is still very limited empirical data on overall mortality during the pandemic based on community data from Africa. Though our trial covers only a narrow time window, it suggests that the impact of COVID-19 on the overall mortality may have been less severe than feared. Hence, as also shown in other African HDSS sites [33,34], the lower death toll in some African settings cannot be fully explained by lack of data.

## Conclusions

Providing free facemasks and messages about their correct use did not substantially increase facemask use in the context of mandated use in urban Guinea-Bissau. While the point estimates for COVID-19-like illness are similar to prior trials, the facemask distribution had no statistically significant effect on the risk of COVID-19-like-illness, consultations or mortality.

## Supporting information

**S1 Text. Supplementary Methods, Supplementary Results, Fig A-J, Tables A-G.**
(DOCX)

**S1 Protocol.**
(PDF)

## Acknowledgments

We sincerely thank all trial participants for their contribution to the trial. We are grateful to the Bandim Health Project data collectors for their efforts in collecting the data. We thank FORCE Technology for testing our facemask material free of charge, Engineers Without Borders Denmark for administrative support. We are indebted to Christine S. Benn for her role in conceptualising the trial and commenting on the results.

## Author Contributions

**Conceptualization:** Line M. Nanque, Andreas M. Jensen, Sebastian Nielsen, Dylan Cawthorne, Kjeld Jensen, Cesario L. Martins, Amabelia Rodrigues, Ane B. Fisker.

**Data curation:** Line M. Nanque, Andreas M. Jensen, Ane B. Fisker.

**Formal analysis:** Line M. Nanque, Sebastian Nielsen, Ane B. Fisker.

**Investigation:** Line M. Nanque, Andreas M. Jensen, Arthur Diness, Justiniano S. D. Martins, Elsi J. C. Ca, Ane B. Fisker.

**Methodology:** Line M. Nanque, Andreas M. Jensen, Amabelia Rodrigues, Ane B. Fisker.

**Project administration:** Carlos Cabral, Ane B. Fisker.

**Software:** Andreas M. Jensen.

**Supervision:** Ane B. Fisker.

**Validation:** Line M. Nanque, Dylan Cawthorne.

**Visualization:** Ane B. Fisker.

**Writing – original draft:** Line M. Nanque, Ane B. Fisker.

**Writing – review & editing:** Line M. Nanque, Andreas M. Jensen, Arthur Diness, Sebastian Nielsen, Carlos Cabral, Dylan Cawthorne, Justiniano S. D. Martins, Elsi J. C. Ca, Kjeld Jensen, Cesario L. Martins, Amabelia Rodrigues, Ane B. Fisker.

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
