## [Decision Letter · Decision Letter 0]

23 Aug 2023

PGPH-D-23-00950

Effect of distributing locally produced cloth facemasks on COVID-19-like illness and all-cause mortality – a Cluster-Randomised Controlled Trial in urban Guinea-Bissau

Dear Dr. Fisker,

Thank you for submitting your manuscript to PLOS Global Public Health. After careful consideration, we feel that it has merit but does not fully meet PLOS Global Public Health’s publication criteria as it currently stands. Therefore, we invite you to submit a revised version of the manuscript that addresses the points raised during the review process.

Please note that we have only been able to secure a single reviewer to assess your manuscript. We are issuing a decision on your manuscript at this point to prevent further delays in the evaluation of your manuscript. Please be aware that the editor who handles your revised manuscript might find it necessary to invite additional reviewers to assess this work once the revised manuscript is submitted. However, we will aim to proceed on the basis of this single review if possible. 

We look forward to receiving your revised manuscript.

Kind regards,

Julia Robinson

Staff Editor

Journal Requirements:

1. Please provide separate figure files in .tif or .eps format only and remove any figures embedded in your manuscript file. Please also ensure all files are under our size limit of 10MB.

Additional Editor Comments (if provided):

Reviewers' comments:

Reviewer's Responses to Questions

**Comments to the Author**

1. Does this manuscript meet PLOS Global Public Health’s publication criteria? Is the manuscript technically sound, and do the data support the conclusions? The manuscript must describe methodologically and ethically rigorous research with conclusions that are appropriately drawn based on the data presented.

Reviewer #1: Yes

2. Has the statistical analysis been performed appropriately and rigorously?

Reviewer #1: I don't know

3. Have the authors made all data underlying the findings in their manuscript fully available (please refer to the Data Availability Statement at the start of the manuscript PDF file)?

Reviewer #1: No

4. Is the manuscript presented in an intelligible fashion and written in standard English?

Reviewer #1: Yes

5. Review Comments to the Author

Reviewer #1: In order to enhance the clarity and transparency of the manuscript, it is recommended to make several improvements.

The notation "(OR=0.83 (0.56-1.24))" should be revised to "(OR=0.83, 95% CI: 0.56-1.24)" instead of using double brackets, as this format is commonly used in reporting confidence intervals.

Moreover, it is important for the manuscript to adhere to the CONSORT (Consolidated Standards of Reporting Trials) statement, which provides guidelines for the transparent reporting of randomized controlled trials. Following the CONSORT statement will help ensure that all essential elements of the study design and methodology are clearly described, facilitating reproducibility and proper evaluation of the research.

To improve the clarity of the methods section, it is necessary to include a clear and concise description of the study design using the PICO (Population, Intervention, Comparator, Outcome) format. This format allows readers to quickly grasp the key components of the study, including the target population, the interventions being investigated, the comparator groups, and the outcomes measured.

In the manuscript, it is crucial to provide specific details regarding the intervention groups and comparator groups, including their enrollment criteria, characteristics, and any relevant demographic information. Clearly stating the enrollment process will ensure transparency and facilitate a better understanding of the study population.

A separate section dedicated to the randomization process should be included, offering comprehensive details on how participants were assigned to intervention or comparator groups.

Assessing and reporting adherence to the interventions is crucial in clinical trials. It is important to mention how adherence was measured and whether there were any instances of non-adherence. If non-adherence occurred, the manuscript should describe how it was addressed, as this can potentially impact the study outcomes.

Missing observations should also be addressed in the manuscript. It is important to clarify whether any observations were missing and describe the method used to handle missing data. Furthermore, the type of analysis performed should be clearly stated, whether it was based on the per-protocol analysis, end-of-study analysis, or intention-to-treat analysis, as this can influence the interpretation of the study results.

Finally, it is essential to assess the balance of outcome measurements between the intervention and comparator groups. If there were any differences in the outcome measurements at baseline, it should be discussed in the manuscript to ensure the validity of the study findings.

6. PLOS authors have the option to publish the peer review history of their article (what does this mean?). If published, this will include your full peer review and any attached files.

**Do you want your identity to be public for this peer review?** For information about this choice, including consent withdrawal, please see our Privacy Policy.

Reviewer #1: **Yes: **Humayun Kabir

---

## [Decision Letter · Decision Letter 1]

12 Dec 2023

PGPH-D-23-00950R1

Effect of distributing locally produced cloth facemasks on COVID-19-like illness and all-cause mortality – a Cluster-Randomised Controlled Trial in urban Guinea-Bissau

Dear Dr. Fisker,

Thank you for submitting your manuscript to PLOS Global Public Health. After careful consideration, we feel that it has merit but does not fully meet PLOS Global Public Health’s publication criteria as it currently stands. Therefore, we invite you to submit a revised version of the manuscript that addresses the points raised during the review process.

We look forward to receiving your revised manuscript.

Kind regards,

Miquel Vall-llosera Camps

Staff Editor

Journal Requirements:

2. We have noticed that you have uploaded Supporting Information files, but you have not included a list 3of legends. Please add a full list of legends for your Supporting Information files after the references list.

Reviewers' comments:

Reviewer's Responses to Questions

**Comments to the Author**

1. If the authors have adequately addressed your comments raised in a previous round of review and you feel that this manuscript is now acceptable for publication, you may indicate that here to bypass the “Comments to the Author” section, enter your conflict of interest statement in the “Confidential to Editor” section, and submit your "Accept" recommendation.

Reviewer #2: (No Response)

2. Does this manuscript meet PLOS Global Public Health’s publication criteria? Is the manuscript technically sound, and do the data support the conclusions? The manuscript must describe methodologically and ethically rigorous research with conclusions that are appropriately drawn based on the data presented.

Reviewer #2: Yes

3. Has the statistical analysis been performed appropriately and rigorously?

Reviewer #2: Yes

4. Have the authors made all data underlying the findings in their manuscript fully available (please refer to the Data Availability Statement at the start of the manuscript PDF file)?

Reviewer #2: No

5. Is the manuscript presented in an intelligible fashion and written in standard English?

Reviewer #2: Yes

6. Review Comments to the Author

Reviewer #2: The authors report the findings of a cluster-randomized trial in which cloth facemasks were distributed and telephone interviews were conducted to assess the impact of COVID-19-like illnesses and mortality over 4 months. The authors should be commended for carrying out such a large-scale trial during the pandemic, using locally-sourced materials to produce face masks. The trial was conducted within the Health and Demographic Surveillance System in Guinea-Bissau, providing a well-enumerated population. The authors performed appropriate statistical tests of their data from which to draw conclusions. Unfortunately, the intervention did not impact the study outcomes. The authors acknowledge several limitations, including lack of impact on behaviors, spillover effects of the intervention, and limited recall. In general, the manuscript is well-written and highlights the numerous challenges of conducting studies during outbreaks.

Major comments:

Lines 105-107: Were all individuals within a household enrolled? Or was one individual selected per household? Please provide additional details around enrollment selection. This is important for the interpretation of the findings.

Line 145-146: Clarify whether “positive test” refers to PCR, antigen, and/or antibody testing.

Lines 157-159: How was the observer able to distinguish study facemasks? Were they clearly distinctive from other facemasks? The photos in the supplement do not show obvious differences.

Lines 344-348: A major limitation of the Danish study was the lack of assessment of compliance with face mask use in the intervention arm. This should be highlighted as an important difference from the Bangladeshi study. Face masks can only be expected to have an effect when they are worn, which is relevant to the current study’s findings.

Given that the study was mired by logistical constraints of doing research during a pandemic, it would be interesting to get the authors’ perspective on how the scientific community could be better situated to generate evidence during future outbreaks, including a brief mention of lessons learned.

Minor comments:

Lines 94-95: It would be helpful to understand more about the mask mandate. Was there any enforcement? When did the mandate end? Was it anticipated to have any effect on the trial results (e.g. improving mask wearing in both the control and intervention groups)?

Lines 95-96: It would be helpful to better understand the curfews that were in place. Did they last for the duration of the study? What were the restrictions? If residents could not leave their houses, masks would not be expected to have an effect. However, this does not seem to be the case since many participants reported interacting with groups of >20 people on a regular basis.

Lines 136-138: It would be helpful to have additional details in the supplement about the tests that were performed to determine the filtration efficiency of the masks. EN DIN 149 is mentioned as the standard, but no citation is provided. For example, what was the flow rate? Was air flow resistance measured? Were measurements assessed after washing and drying the facemasks as this can significantly affect the fabric performance? Were multiple samples of plain-weave cotton fabric tested? Filtration results can vary greatly by thread count. How was this controlled for across production? How were sizes determined for study participants? Were there any acceptability studies?

Lines 150-151: Soliciting information from family members and neighbors when participants were absent may have resulted in recall bias, though presumably this should have been similar between the control and intervention groups.

Table 1: For number of people sleeping in the same room, how can the p-value be 0.001 if the median and IQR were the same in the control and intervention arms?

Recall seems to be an issue given the much higher reporting among those who were contacted at 2-month intervals compared with study end only (Lines 287-290 and 293-295).

Line 299: Follow-up is duplicated.

7. PLOS authors have the option to publish the peer review history of their article (what does this mean?). If published, this will include your full peer review and any attached files.

**Do you want your identity to be public for this peer review?** For information about this choice, including consent withdrawal, please see our Privacy Policy.

Reviewer #2: No

---

## [Editor Report · Decision Letter 2]

22 Jan 2024

Effect of distributing locally produced cloth facemasks on COVID-19-like illness and all-cause mortality – a Cluster-Randomised Controlled Trial in urban Guinea-Bissau

PGPH-D-23-00950R2

Dear Dr. Fisker,

We are pleased to inform you that your manuscript 'Effect of distributing locally produced cloth facemasks on COVID-19-like illness and all-cause mortality – a Cluster-Randomised Controlled Trial in urban Guinea-Bissau' has been provisionally accepted for publication in PLOS Global Public Health.

Best regards,

Julia Robinson

Executive Editor